# Unsupervised mapping of phase diagrams of 2D systems from infinite projected entangled-pair states via deep anomaly detection

Korbinian Kottmann[1*], Philippe Corboz[2], Maciej Lewenstein[1,3], Antonio Acín[1,3]

**1** ICFO - Institut de Ciencies Fotoniques, The Barcelona Institute of Science and Technology, Av. Carl Friedrich Gauss 3, 08860 Castelldefels (Barcelona), Spain
**2** Institute for Theoretical Physics and Delta Institute for Theoretical Physics, University of Amsterdam, Science Park 904, 1098 XH Amsterdam, The Netherlands
**3** ICREA-Institucio Catalana de Recerca i Estudis Avançats, Lluis Companys 23, 08010 Barcelona, Spain
* KorbinianKottmann@icfo.eu

July 9, 2021

## Abstract

We demonstrate how to map out the phase diagram of a two dimensional quantum many body system with no prior physical knowledge by applying deep *anomaly detection* to ground states from infinite projected entangled pair state simulations. As a benchmark, the phase diagram of the 2D frustrated bilayer Heisenberg model is analyzed, which exhibits a second-order and two first-order quantum phase transitions. We show that in order to get a good qualitative picture of the transition lines, it suffices to use data from the cost-efficient simple update optimization. Results are further improved by post-selecting ground-states based on their energy at the cost of contracting the tensor network once. Moreover, we show that the mantra of "more training data leads to better results" is not true for the learning task at hand and that, in principle, one training example suffices for this learning task. This puts the necessity of neural network optimizations for these learning tasks in question and we show that, at least for the model and data at hand, a simple geometric analysis suffices.

# 1 Introduction

With the introduction of a new data-driven computation paradigm, machine learning (ML) techniques have been very successful in performing recognition tasks and have had a big impact on industry and society. ML has been successfully applied to a variety of physical problems, and vice versa, physics has inspired new directions to explore in understanding or improving ML techniques [1]. Among the most prominent and successful applications of ML in physics is the classification of phases in many body physics [2–19]. Of particular interest are unsupervised methods that require no or little prior information for labeling [2,3,6–8,20]. In particular, phase diagrams have been mapped out in a completely unsupervised fashion with no prior physical knowledge from 1D tensor network data [21] and experimental data [22] via *anomaly detection.*

In this work, we extend the application of anomaly detection for phase characterization to 2D quantum many body systems with projected entangled-pair states (PEPS). PEPS have been introduced as an efficient ansatz for ground states for 2D Hamiltonians [23–25] and extended to simulate the thermodynamic limit with infinite PEPS (iPEPS) [26]. Various methods to optimize the iPEPS tensors exist, including (fast-) full update imaginary time evolution algorithms [26, 27] and energy minimization approaches [28–30]. Computationally cheaper alternatives were introduced with the simple update [31, 32] and cluster update [33] algorithms that perform optimizations locally at the cost of numerical accuracy. Progress in the systematic study of continuous phase transitions has recently been achieved based on a finite correlation length scaling analysis [34–36].

The intention of the scheme presented in this work is not to improve numerical accuracy of determining phase boundaries, but to obtain a qualitative phase diagram with *low computational cost* and *no physical a priori knowledge.* The former, low computational cost, is achieved by employing the simple update optimization algorithm with contractions omitted throughout the whole process. Physical knowledge in this scheme is redundant as we resort to quantities obtained directly from the iPEPS wave function from simulations as inputs for the machine learning method; in this case singular values between bonds or reduced density matrices. In other words, we do not need to choose and compute suitable observables that contain sufficient information about the phase boundaries for the machine learning processing. In 1D systems, the singular values between bonds have a clear physical interpretation, as they characterize the entanglement properties between the subsystems at each end of the bond. In 2D, however, there is no such interpretation and it is non-trivial to show that singular values at bonds are still sufficient to determine phase boundaries. In the considered approach, phase boundaries are characterized without the need to know the order parameter or symmetry groups of the phases. In fact, in a scenario where we are given data from iPEPS to analyze, in principle we do not even need to know the Hamiltonian.

In contrast to supervised methods, where at least a rough idea of the regions of different phases (and the number of separate phases) is needed for labeling a training set, we do not need to know anything about the phase diagram by using unsupervised anomaly detection.

This is because in this scheme a region of the diagram is chosen to represent normal data and is then tested for the whole diagram. Initially, this normal region is chosen randomly and may contain states from one or multiple phases. When states from different phases than it has been trained on are tested, they are marked as anomalies. Between those states and the training region, there is a transition from normal to anomalous data that corresponds to the phase transition. In the next training iteration, the normal region is put where anomalies have been found in the previous round and the process repeated until no previously unseen anomalous region is found. This process only needs $\mathcal{O}(N_{\mathrm{phases}})$ iterations where $N_{\mathrm{phases}}$ is the number of phases in the diagram. This is in contrast to *learning by confusion* schemes where the phase diagram is scanned by iteratively shifting the labeling and retraining [3,6].

In spirit, the approach presented here is similar to the method described in [37,38]. There, phase transitions are determined by looking at the overlap (fidelity) between neighboring ground states in the phase diagram, with a drop in the fidelity at quantum phase transitions as the overlap between states from different phases is small (zero) for finite (infinite) system sizes. The big advantage of the approach presented here is that we avoid computationally expensive contractions of the tensor network, which, in contrast, is needed to compute overlaps.

As an example, we examine the 2D frustrated bilayer Heisenberg model, a challenging problem which suffers from the negative sign problem [39]. The model contains two 1st order and one 2nd order phase transition and is therefore a good benchmark for the success of this method. This manuscript is organized as follows: The general approach of applying anomaly detection with neural networks to map out phase diagrams is described in section 2. Details on iPEPS are described in section 3 and a brief overview of the 2D frustrated bilayer Heisenberg model is given in section 4. The results are then presented in section 5, followed by our conclusions in section 6.

## 2  Anomaly Detection

We follow the approach in [21, 22], where it was shown that phase diagrams can be mapped out from different data types in an unsupervised fashion via *anomaly detection*. The scheme works in the following way: We employ a special neural network architecture, called an autoencoder, to efficiently decode and encode data of the type it has been trained on (data specific compression). For the training[1], we define a training dataset containing *normal* data. The autoencoder is trained to efficiently reproduce data with the same or similar characteristics. Anomalies are detected by deviations of a loss function between input and output of the autoencoder, compared to the region it has been trained on and amount to separate phases in the diagram. This training has to be performed only $\mathcal{O}(N_{\mathrm{phases}})$ times where $N_{\mathrm{phases}}$ is the number of phases present in the phase diagram. Moreover, this procedure does not necessitate any prior physical knowledge about the system as one starts with an arbitrary parameter range, typically at the origin of the parameter space. From there, abrupt changes in the reproduction loss are saved as possible phase boundaries and the next training iteration is done in the region with the highest loss after the previous training. Note that in principle one does not even need to know the underlying Hamiltonian, it suffices to be provided with data and the corresponding physical parameters.

---

[1] *Training* in machine learning refers to data-specific optimization. This is described in more detail below around eq. (1).

We employ autoencoder neural network architectures implemented with TensorFlow [40]. An autoencoder is composed of an encoder and a decoder. The encoder takes the input $x$ and maps it to a latent space variable $z$. This latent space variable is then mapped by the decoder to the output $y(x)$. Both encoder and decoder are composed of multiple consecutive layers, parametrized by free parameters $\theta$. We tried different architectures comprising different combinations of fully-connected and convolutional layers. We find no specific model dependence, with different architectures performing similarly such that simple *vanilla* autoencoders composed solely of fully-connected layers suffice and are used throughout this paper. For details about the implementation see [41]. The goal of the autoencoder is to reproduce $x$, i.e. matching the output of the network with its input $y(x) = x$, which is achieved by minimizing the reproduction loss

$$L(\theta) = \sum_i ||x_i - y_i(x_i)||^2 \tag{1}$$

with respect to the free parameters $\theta$ ($y(x)$ implicitly depends on $\theta$). The sum reaches over the training examples defined for the training iteration. Here we have chosen the loss to be the mean squared error as it is simple and effective for our task, but in principle there is a variety of possible and valid loss functions depending on the task and data at hand. The optimization task of minimizing $L$ is achieved by gradient descent $\theta \mapsto \theta - \alpha \nabla_\theta L(\theta)$, i.e. computing the gradient of $L$ and changing the free parameters in the opposite direction for some stepsize $\alpha$ (hyper parameter given by the user). For neural networks, there is an efficient implementation called backpropagation [42]. We employ ADAM, a modern optimization scheme with adaptive stepsizes based on gradient descent with backpropagation for faster optimization [43].

## 3 Infinite projected entangled-pair states

An iPEPS [26] is a tensor network ansatz to represent 2D ground states directly in the thermodynamic limit and can be seen as a generalization of 1D infinite matrix product states (iMPS) to 2D. The ansatz consists of a unit cell of rank-5 tensors repeated periodically on a lattice. Here we use a unit cell with two tensors arranged in a checkerboard pattern on a square lattice, with one tensor per dimer in the bilayer model introduced in Sec. 4. Each tensor has one physical index representing the local Hilbert space of a dimer and four auxiliary indices, which connect to the four nearest-neighbor tensors. The accuracy of the variational ansatz is systematically controlled by the bond dimension $D$ of the auxiliary indices. To improve the efficiency of the calculation we use tensors which exploit the U(1) symmetry of the model [44, 45].

In this paper the optimization of the tensors is done based on an imaginary time evolution, which involves a truncation of a bond index at each time step (see Refs. [26, 27, 46] for details). While an optimal truncation requires a full contraction of the 2D tensor network (called the full update [26]), which is computationally expensive, there exist also local, approximate schemes avoiding the full contraction. In the simple-update approach [31, 32], which we use in this work, the truncation is performed by a local singular value decomposition of two neighboring tensors. A more accurate, but still local scheme is provided by the cluster update introduced in Ref. [33], in which the truncation is done based on a cluster of tensors, where the accuracy is controlled by the cluster size.

We start the iPEPS optimizations from random initial states, thereby avoiding the need for any knowledge of the system a priori. Depending on the initial state, the iPEPS may converge to a local minimum. To improve the convergence behavior, the state is initially evolved for a few steps at large bond dimension and large imaginary time step, then projected to $D = 1$, and then evolved at the target $D$. Optimization runs are discarded and repeated when convergence is not reached after a certain number of steps. Finally, the state is further evolved at the target dimension at a smaller time step until convergence is reached. We found that this scheme improves the efficiency and quality of the results, compared to an evolution only at the target $D$, especially close to the first order phase transition line of the model.

As input data for the anomaly detection, we use the singular values of the four auxiliary bonds obtained from the simple-update approach. In 1D iMPSs in canonical form, they correspond to the Schmidt coefficients, characterizing the entanglement between the two sides of the system connected by the bond. In 2D, however, there exists no canonical form and the singular values do not correspond to the Schmidt coefficients because of the loops in the tensor network ansatz. Still, we will show that the singular values contain information that can be used for and interpreted by the machine learning algorithm to characterize the underlying ground state.

For comparison we also consider the 2-site reduced density matrix as input data, which is computed by contracting the 2D tensor network using the corner transfer matrix method [47–49].

# 4  Model

We test the anomaly detection scheme in combination with iPEPS for the $S = 1/2$ 2D frustrated bilayer Heisenberg model - a challenging problem, where a large part of the phase diagram is inaccessible to Quantum Monte Carlo due to the negative sign problem [39]. The model can be represented as a two-dimensional lattice of coupled dimers, formed by the two $S = 1/2$'s of the adjacent layers. The Hamiltonian reads

$$H = \sum_i J_\perp \vec{S}_{i,1} \cdot \vec{S}_{i,2} + \sum_{\substack{i,m=1,2 \\ j=i+\hat{x},i+\hat{y}}} \left[ J_\parallel \vec{S}_{i,m} \cdot \vec{S}_{j,m} + J_x \vec{S}_{i,m} \cdot \vec{S}_{j,\bar{m}} \right] \tag{2}$$

where $J_\parallel$ is the nearest-neighbor intralayer coupling and $J_\perp$ ($J_x$) the nearest (next-nearest) neighbor interlayer coupling, $i$ is the index of a dimer, $j$ runs over the nearest-neighbor dimers, $m$ denotes the two layers, and $\bar{m}$ the layer opposite to $m$.

In the limit of strong $J_\perp$ the ground state is a dimer singlet (DS) state with vanishing local magnetic moment. For $J_x = 0$ the model is unfrustrated with an ordered bilayer antiferromagnetic (BAF) ground state for $J_\perp/J_\parallel < 2.5220(2)$ [50], separated from the DS phase by a continuous transition. The limit $J_x = J_\parallel$ corresponds to the fully-frustrated Heisenberg bilayer model with a dimer-triplet antiferromagnetic (DTAF) ground state for $J_\perp/J_\parallel < 2.3279(1)$ [51], in which spins on a dimer are parallel (in contrast to the BAF phase where the spins on a dimer are antiparallel). The ground state phase diagram of the full model was mapped out with iPEPS in Ref. [39] and is shown in fig. 1 1a) by the white-dashed lines. It hosts a quantum critical endpoint at which the line of continuous transitions between the BAF and DS phase terminates on the first order line separating the DTAF phase from the DS and BAF phases.

# 5   Numerical Results

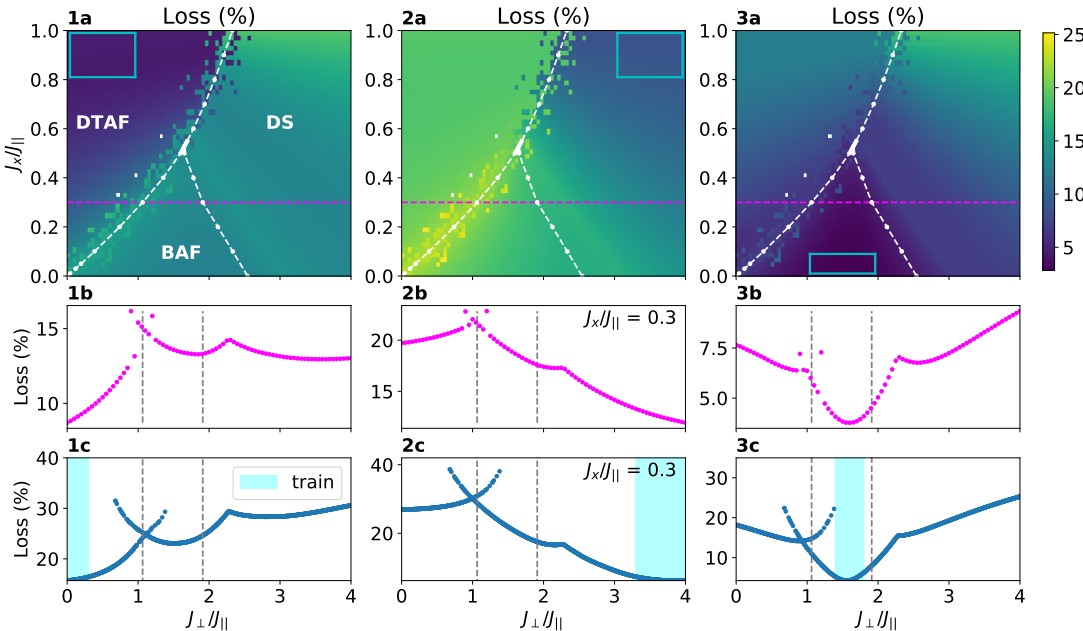

Figure 1: Three training iterations to map out all three phases of the phase diagram without contraction and prior knowledge of the system, i.e. using simple update algorithm starting from random initial iPEPS. The cyan rectangles show the training regions. Overlayed in white are theoretical predictions, extrapolated to the infinite $D$ limit with full update optimization from [39]. Deviations of the second order BAF-DS transition are expected due to finite $D = 6$ and simple update optimization. 1a) Starting the training at the top-left (in DTAF phase) yields the first-order transition line. Even the second order transition line is already pronounced inside the region of higher loss beyond the first order line. 2a) Second iteration in the region of highest loss (DS phase) from the previous picture showing the part of the first order line adjacent to the DS phase and the second order line. 3a) Confirming and completing the picture by training in the BAF phase. The second row (1b-3b) is a single cut as indicated by the magenta line in the phase diagram above. The third row (1c-3c) shows the loss after training and evaluating extra single cut data with five independent simulations per data point. Around the first order transition two branches are obtained due to the characteristic hysteresis behavior.

We now use the anomaly detection scheme described in section 2 with data from iPEPS, described in section 3 to map out the phase diagram of the model without prior physical knowledge in fig. 1. Three training iterations suffice to map out the boundaries of all three phases of the system. We start in the top-left corner of the phase diagram corresponding to the DTAF phase in fig. 1 1a) and obtain the first order transition line. The noise around the line is due to hysteresis effects in the vicinity of the first order phase transition: Depending on the random initial tensors, the converged states end up in one of the two adjacent phases. We will later see how a sharp transition line can be obtained by measuring the energy of the states. Note how the second order transition line is already indicated within the anomalous

region of high loss. The second training is performed in the region of highest loss from the previous iteration in the top-right corner corresponding to DS in fig. 1 2a). The second order transition line to the BAF phase is again signaled by a bump in the loss diagram. To complete and confirm both lines, training is performed in the BAF phase in fig. 1 3a). In fig. 1 (1c-3c) we present data for the single cut at $J_x/J_{||} = 0.3$ with five independent simulations for each value of $J_\perp/J_{||}$, illustrating the characteristic hysteresis behavior around the first order DTAF-BAF transition.

All the results in fig. 1 are overlaid with the previous iPEPS simulation results from [39]. Note that those results are much more precise since the iPEPS were optimized with the more expensive full update algorithm and the data has been extrapolated to the infinite $D$ limit, whereas here we only consider simple-update data at finite bond dimension $D = 6$. Thus a quantitative deviation can be expected, especially for the location of the second order BAF-DS transition line. However, the main goal here is to get a cheap and fast overview of the phase diagram, which serves as a useful starting point for more accurate numerical investigations (e.g. based on a finite correlation length scaling analysis [34, 35]). We note that the ML approach could also be combined with the more accurate cluster update scheme [33], which is still a local approach, or with the full-update [26, 27] or energy minimization schemes [28–30, 52], which require full contractions. [2]

To get an even clearer picture of the predicted phase boundaries, we compute the energy of the states to post-select the best ground states. The ground-state optimization is initialized from three different initial iPEPS (a representative point in each phase) and only the lowest in energy is kept. The boundaries in the resulting pictures are now much more pronounced at the cost of invoking contractions to calculate the energy, yet still not taking any physical knowledge of the order into account. For the sake of showcasing the method with different inputs, we here use the 2-site reduced density matrix $\rho^{(2)}$ but also confirm again the viability with singular values.

We proceed in an analogous fashion and perform three trainings to map out the phase diagram in fig. 2. The phase boundaries appear even sharper compared to fig. 1, especially for the first order transition line with a corresponding discontinuity (jump) in loss at the transition. In fig. 2 c) we confirm that the results are qualitatively the same when using the singular values as an input instead.

We note that, while in practice the singular values are found to be gauge-invariant (see also Ref. [53]), the reduced density matrix is not necessarily unique. If the state breaks, e.g., SU(2) spin symmetry, then different random initial states will lead to different reduced density matrices (since the local magnetic moments can be aligned along different directions). In Fig. fig. 2, this is not an issue, because each anomaly detection is based on a single initial state (which fixes the direction of the magnetic moments), and by using U(1) symmetric tensors, the magnetic moments are automatically parallel to the z-axis. Alternatively, one can also consider the eigenvalue spectrum of the reduced density matrix as input data, which is gauge-invariant, as shown in fig. 2 (1c-3c).

It is a common mantra in machine learning that more data always yields better results. In the present study, where the machine learning task is to find the phase boundary from singular values, we find this to actually not be the case. Also, the result of the algorithm is not sensitive to the extent of the training region, i.e. how far in parameter space the examples during training reach. It seems that one example of the phase already captures

---

[2]In the latter two cases the singular values can be extracted using the approach described in Ref. [53].

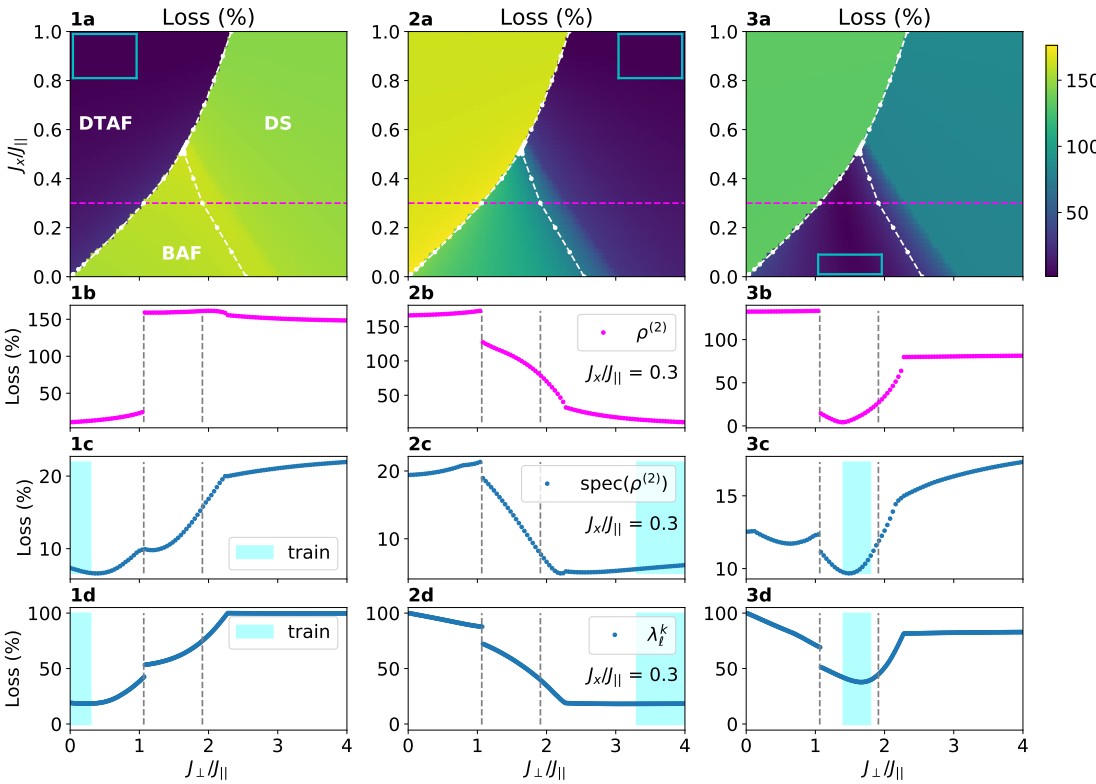

Figure 2: Three training iterations to map out all three phases of the phase diagram. Here, the reduced 2-site density matrix is used as input data. In comparison to fig. 1, the first order transition line is much sharper as the ground states were post-selected from energy considerations. The second row (b) shows the line at $J_\perp/J_{||} = 0.3$ as indicated by the dotted magenta line in row (a). In row (c), the eigenvalues of the reduced 2-site density matrix in log-scale are used. The training is done just for the single cut at $J_\perp/J_{||} = 0.3$. Row (d) uses again the singular values like in fig. 1.

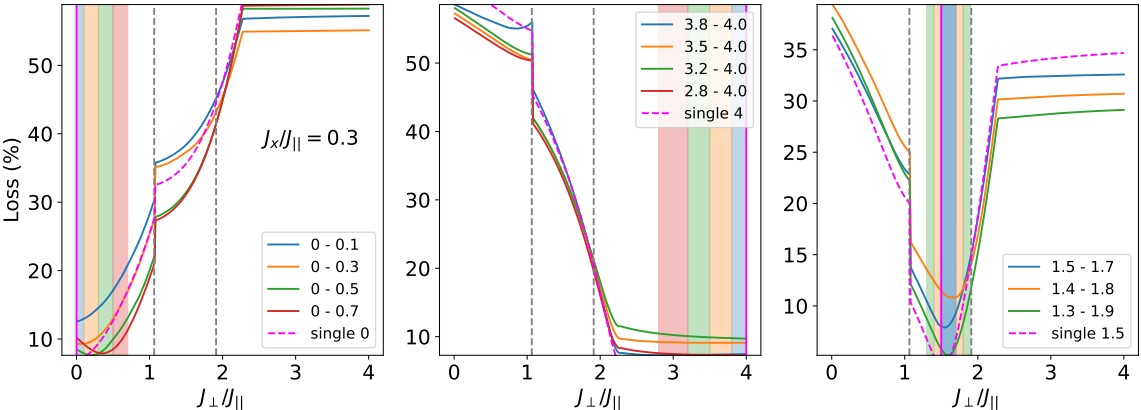

Figure 3: Varying the training region to show that the outcome of the algorithm is not sensitive to the number of training examples $N_{\text{ex.}}$ and the extent of the training region. The number of epochs $N_{\text{epochs}}$ is chosen such that $N_{\text{epochs}} \cdot N_{\text{ex.}} = \text{const.}$, i.e. the neural network *sees* the same amount of examples throughout all training iterations. We see that we can even map out the regions with just one single training example (dotted magenta curve).

the characteristic features and that the data within the phase is so homogeneous that adding more examples does not improve the result. To show this, we take a single cut at $J_x/J_{||} = 0.3$ and vary the extent of the training regions in fig. 3 for singular value data with $D = 10$. In all cases the predicted transitions are the same and the result is insensitive to the chosen training region. We can put this to the extreme and only use one single training example $N_{\text{ex.}} = 1$ and still obtain the same results. In all the cases of training regions in fig. 3, the number of epochs $N_{\text{epochs}}$, that is the number of times the neural network processes the full training set, is chosen in such a way that $N_{\text{epochs}} \cdot N_{\text{ex.}}$ is held constant, such that during training the same number of examples are processed for a fair comparison.

This raises the question of the necessity of the neural network machinery for the anomaly detection. In fig. 4 we show simple, purely geometric and data-driven approaches that indicate the phase boundaries in the spirit of anomaly detection without using neural networks. In the first case, we compute the inner product between normalized singular value vectors $s_i$ for different physical parameters. Here, the inner product is just the *standard* inner vector product

$$\text{inner}(s_i, s_j) = \sum_k s_i^k s_j^k. \tag{3}$$

The normalization is done such that $\text{inner}(s_i, s_i) = 1$. Using inner products, there is a clear interpretation of the overlap values and we can see that the contrast in fig. 4 1) is of order $\sim 0.01$ and therefore arguably small. We get better results when using a geometric similarity measure

$$\text{similarity}(s_i, s_j) = \sum_k |s_i^k - s_j^k|^2 \tag{4}$$

between a fixed normalized singular value vector along the physical parameter space in fig. 4 2). Note that this is equivalent to the loss in eq. (1), used for the autoencoder. These results are now very similar to the ones obtained with the autoencoder in fig. 3. An interesting open

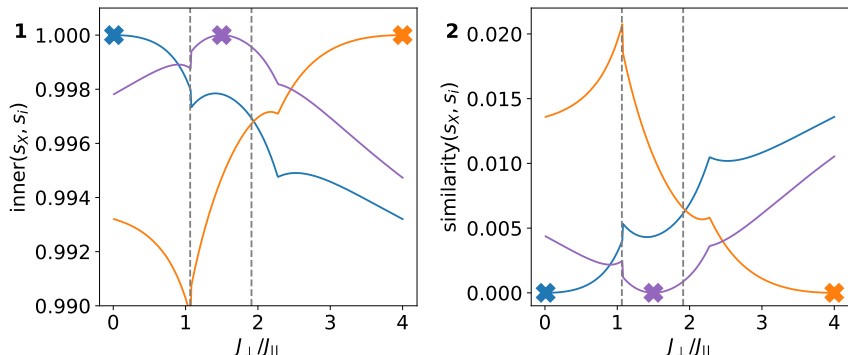

Figure 4: Detecting the phase boundaries with purely geometric data driven methods at $J_x/J_{||} = 0.3$. 1) Inner product between the singular values $S_i$ along the parameter space and a fixed point (indicated by the X). 2) Geometric similarity measure equivalent of the loss for the autoencoder eq. (1).

question to answer in future work is whether such a data-driven geometric analysis in the spirit of machine learning but without neural networks suffice in general or if this is specific to the model and data at hand.

## 6  Conclusions

In this work, we showed how to combine anomaly detection, a method for unsupervised ML, with iPEPS, a tensor-network ansatz for variational optimization, to map the phase diagram of 2D systems. By employing ML, we circumvented the necessity for defining and calculating suitable observables to identify the phases. Furthermore, no prior physical knowledge was required to run the unsupervised anomaly detection (i.e. no labels needed). We saw that a successful training can be achieved with an arbitrarily small amount of examples, therefore making the amount of data generated a matter of aesthetics by the user. Based on this, we saw that for the present model and data, purely geometric and data-driven analyses sufficed and raised the question whether such approaches are feasible in general for finding phase transitions from data.

It shall also be mentioned that the dimension of the data being used was small, $D \times 4$ in the case of the singular values, such that in this case there was no necessity for dedicated machine learning hardware like graphical processing units (GPUs) and all trainings were performed in less than 10 seconds on a commercial laptop with an Intel i7-4712HQ CPU, see [41]. Here we used the resource economic simple update algorithm to obtain the iPEPS ground states, but we note that the ML approach could also be combined with more accurate (but computationally more expensive) optimization approaches. In summary, we provided a very fast and efficient approach to qualitatively map out the phase diagram of 2D systems with no prior physical knowledge of the underlying system, offering a powerful way to obtain quick insights into the physics of new models.

# Acknowledgements

We thank T. Neupert and E. van Nieuwenburg for useful suggestions. This work was supported by the European Union's Horizon 2020 research and innovation programme under the Marie Sklodowska-Curie grant agreement No 713729 (K.K.), and grant agreement No 677061 (P.C.), the ERC AdG's NOQIA and CERQUTE, Spanish MINECO (FIDEUA PID2019-106901GB-I00/10.13039 / 501100011033, FIS2020-TRANQI, Severo Ochoa CEX2019-000910-S and Retos Quspin), the Generalitat de Catalunya (CERCA Program, SGR 1341, SGR 1381 and QuantumCAT), Fundacio Privada Cellex and Fundacio Mir-Puig, MINECO-EU QUANTERA MAQS (funded by State Research Agency (AEI) PCI2019-111828-2 / 10.13039/501100011033), EU Horizon 2020 FET-OPEN OPTOLogic (Grant No 899794), and the National Science Centre, Poland-Symfonia Grant No. 2016/20/W/ST4/00314, Marie Sklodowska-Curie grant STRETCH No 101029393.

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
