# Peer review of "Unsupervised mapping of phase diagrams of 2D systems from infinite projected entangled-pair states via deep anomaly detection"

_SciPost Physics_

## Round 1 · Referee Report · Everard van Nieuwenburg (Referee 1) · 2021-6-8

Strengths

1 - Can cheaply map out a phase diagram using an autoencoder trained on a single sample
2 - Physics-agnostic; "just" a dataset (as a function of parameters) suffices
3 - The benchmark model is frustrated, and has both second- and first-order phase transitions.
4 - The provided code is very accessible, and can directly be used to reproduce the results.

Weaknesses

  1. There were a few points at which the manuscript can sometimes leave things unexplained, or where it makes use of concepts before they have been introduced. For example:
  2. The reproduction loss is used before it is introduced
  3. The 'training' of a network on the normal data is used before introduced
  4. It is unclear whether the normal regions for each iteration are chosen manually, or automatically (I had to look at the code to see that they are hardcoded)
  5. The authors write: "When states from different phases than it has been trained on are tested, they are marked as anomalies. In between, there is a transition from normal to anomalous data that corresponds to the phase transition". What does 'in between' here mean? (this becomes clear only later)

  6. What about convergence? For Fig 3 The training was done for a fixed number of epochs, s.t. Num_epochs * Num_samples = const = 4000, for Figs 1 and 2 the training was done over 100 epochs (from the code); the loss has not yet converged, and increasing the training time actually improves the results (e.g. Fig 1 3b has a more pronounced divergence for 600 epochs).

Report

Dear authors,

[Summary] Your manuscript has a clear message, in that it answers the following question affirmatively: "Can the singular values (or the 2-site reduced density matrix) of an iPEPS be used to determine phase boundaries?". You show that using the unsupervised anomaly detection technique, based on autoencoders, the phase diagram of an interesting benchmark system can be qualitatively & cheaply obtained.

[Validity] I believe the results are valid, judging from the manuscript as well as from playing around with the provided source code.

[Originality] The work fits in very nicely with previous literature (especially [21]), combining the technique of 'anomaly detection' with singular values from tensor networks as input (though in previous literature those came from MPSs, for which they have an interpretation). It is promising to see that for the 2D case they still contain information about phase transitions. What I find especially significant is the result that a single set of (6) singular values is sufficient to detect the phase transitions (also see point 3 below+ attached figure).

I have a few questions/comments/suggestions, but have no reservations in recommending this manuscript for publishing in SciPost.

"The ground-state optimization is initialized from three different initial states (a representative state in each phase)..." Do the authors mean a representative point in each phase? Otherwise, a little bit of physical intuition is being put in for the data generation.

2.

It is a common mantra in machine learning that more data always yields better results. Here, this is actually not the case. I agree with the authors that this is indeed a sentiment that exists. In my experience, it typically refers to having more data in the sense of more samples, which for non-physics data often means a larger 'variety' (in the extreme case, more pictures of dogs likely means different dogs and different scenarios; in the physics case, for a fixed parameter window more samples means more of the same). In Fig 3, this is still visible in that the largest windows seem to achieve a lower loss in the anomalous regions, indicating that perhaps the autoencoder was better able to extract a model (which incidentally is not what the authors are after, per se, since they rely on the autoencoder having a large loss to find anomalous regions).

  1. The fact that the authors can detect the anomalies using a single example is very interesting in my opinion. Training an autoencoder on a single datapoint is like building a compression algorithm for 1 specific file. That this compressor works ok-ish for similar files (same phase), and not so much for others (other phase) is very believable, and may be a separate method worthy of further exploration. In fact, I wondered if this means that the autoencoder is superfluous. Just choosing a simple 'distance' metric (y_test = abs(x_test - x_single_sample)) as similarity between the singular value inputs already produces very similar results indeed (see attached figure).

  2. For a while I thought that Figure 1 and Figure 2 seemed to have incorrectly labeled x-axes for rows a) and b), but I realized that they are titles instead. Maybe consider having that in the caption or as an in-plot text instead (as is done in Fig 3)?

  3. Though I am flattered by the double citation, it seems Refs 3 and 37 are identical. At the location where you currently cite [37], I would also draw your attention to your Ref [6], which improves upon the confusion scheme in that it is a more directed search rather than an iterative shifting. Your main point there is still valid, however.

  4. In Fig 3, the same vertical dashed lines for the transitions points would be helpful, there seems to be just a little variation in the prediction points (but well within 'qualitatively the same' phase diagram).

Requested changes

I do not have specific points to change, but would ask the authors to take a look at the points I listed in the report. If they agree, please consider implementing relevant changes.

Attachment

---

## Round 1 · Referee Report · Titus Neupert (Referee 2) · 2021-6-15

Strengths

1- Accessibly and very well written, without too much lingo 2- Methods, parameters, etc. are clearly spelled out 3- Uses an unsupervised method that is also applicable to other data sets with continuous labels. 4- Computational efficiency for the case studied

Weaknesses

1- The principle methodology is not new, but has previously been applied to other numerically obtained (and experimental) data in Refs. 21, 22. Thus, the novel aspects that this work brings in my view are mainly relevant for the tensor network community, but not beyond. 2- No new physical insights have been learned in/enabled by this work as far as I understand.

Report

I enjoyed reading the manuscript as it nicely builds up on previous work, is yet self-contained and explores a very suitable use-case of anomaly detection.

My main question is how "well defined" the algorithm is in this setting. The question arises from ignorance about iPEPS on my side. In principle one should only look at gauge invariant input data for such an algorithm to work or one should make sure that input data is statistically randomized over all gauge-equivalent representations. The problem I am alluding to can be seen most clearly when thinking about quantum mechanical wave functions as input data for an ML algorithm. Clearly the phase of the wave function does not carry any physical information, but depending on the way the data is generated, the phase could vary smoothly over some region in the phase diagram and then abruptly change. The anomaly detection may pick that up as a phase transition, while the physical state has not changed. The same is true for a reduced density matrix. In contrast, a sorted entanglement spectrum is a gauge-invariant quantity. My question is thus: Is the data (singular values of the auxiliary bonds) gauge invariant? In one case, the reduced density matrix is used directly. How is this gauge freedom addressed in this case?

As a second point, I would like to challenge the statement that the "mantra 'more training data leads to better results' is not true". Is this actually meant with respect to the energy post-selection or with respect to the training with a single example? In my understanding, the challenging problems that we address with machine learning are high-dimensional inference. For these, more data is better since it allows for a better approximation of the probability distribution in question. If one finds the contrary, I would claim one is doing something trivial or wrong. For the case at hand: how much worse would one do if one just plots the overlap (some inner product) of the appropriately normalized data with that of these single reference states?

I would be curious to hear the answers to these two questions to be convinced that the method is well-defined and the results are truly nontrivial.

Requested changes

Trivial change requests:

1- References 3 and 37 are the same. 2- y(x) is not in math mode below Eq. 1 3- Sometimes in the writing it is a bit ambiguous in my view whether "initial states" and similar terminology refers to initial states of the neural network or initial states of the tensor network. Especially for readers outside the field a bit more redundant formulation would help. 4- I felt a bit lost with the term 'anomaly detection' until it is nicely explained, not knowing whether I should be familiar with this terminology or not. Maybe making it italic or indicating that an explanation comes up later in the text helps the reader to be patient and read over it.

  • validity: good
  • significance: good
  • originality: high
  • clarity: top
  • formatting: excellent
  • grammar: perfect

Author:  Korbinian Kottmann  on 2021-07-09  [id 1558]

(in reply to Report 2 by Titus Neupert on 2021-06-15)
Category:
answer to question

Concerning the first paragraph of the report:
We thank the referee for raising this interesting and important point. Although, to our knowledge, there is no proof that in iPEPS the simple-update singular values are unique. However, what has been observed in practice so far is that they are indeed invariant (see also Ref. [54] where the singular values are retrieved starting from a standard PEPS representation). Thus, in practice we do not expect any issues when using the singular values as input data.
However, we agree with the referee that using the reduced density matrix (RDM) as input data can be problematic in certain cases. If the ground state breaks a symmetry, then the RDM is no longer unique, but different random initial states may lead to different RDMs (e.g. in case of spontaneous SU(2) spin symmetry breaking, the local magnetic moment may point in different directions depending on the initial state, which is encoded in different RDMs). In our results in Figure 2 this problem does not arise, because the optimization is done starting from a single initial state for each training region (which fixes the direction of the magnetic moments), and by using tensors with a U(1) symmetry the magnetic moments are automatically parallel to the z-axis. We included a discussion of this point in Sec.5 in the revised version. Additionally, as an alternative to the "bare" RDM we also consider the eigenvalue spectrum of the RDM, which is invariant, and present results in an additional row of plots in Fig.2 (1c-3c).

Concerning the second paragraph of the report:
We agree with the referee that most of the groundbreaking applications of ML were made possible due to the availability of bigger datasets and capabilities to process them, and that in general one of the strong suits of ML is high dimensional inference. We do not mean to object to this in general but rather make the point that for the learning task at hand, i.e. learning phase boundaries from (almost) ideal simulation data, we find that this is not the case. The states and corresponding data in the individual phases are so homogeneous that adding more training examples does not significantly improve the results as characteristic features can be learned from just one example. We updated the text here and specified this more clearly to avoid confusion.
We address the second point made by the referee in the general author comment in resubmission, since this is in reply to both referees.

---

## Round 2 · Referee Report · Titus Neupert · 2021-7-9

Report

I think the authors addressed my concerns and comments comprehensively in their reply and I have no further reservations against the publication of this manuscript.

---

## Round 2 · Referee Report · Everard van Nieuwenburg · 2021-7-9

Report

The authors have addressed the points I previously raised, and have in particular included a new figure (Fig. 4) in which they tested a suggestion. I have no further comments on the manuscript as-is, and am happy to recommend it for publishing in SciPost Physics.

---

## Round 2 · Author Response

Both referees pointed out that because the training can be done with just one example, the autoencoder might be superfluous. Referee 1 suggested using a similarity measure equivalent to the loss function for the autoencoder and referee 2 suggested using inner products. So, if we understand correctly, the idea is to perform a kind of data-driven geometric analysis in the spirit of machine learning but w/o machine learning (i.e. neural networks). There has been work done in similar directions, e.g., Ref. [38] showed that phase boundaries can also be determined via inner products of quantum states. One point that we made in the paper is that inner products between quantum states are in fact expensive for 2D tensor networks and can be avoided with our proposed method.

If we understood correctly, the referees raise the very interesting point that overlaps (or similarities) could also be computed from the reduced data that is used for the machine learning protocol. Indeed we find that this is possible and leads to comparable results in quality. For the inner product it is a bit hard to argue as we see that the contrast is arguably small on a range of 0.01 (between 1.00 and 0.99). But in both cases, undeniably, the results are qualitatively comparable. We find this to be very curious and allowed ourselves to add it to the manuscript (Fig. 4).

The beauty of ML methods, like the anomaly detection scheme in discussion here, is that it is a very general framework that is capable of adapting to a data-specific problem by having an over-parametrized objective function that is optimized for the given data. This is in general very powerful as it is very flexible in the problems it can be applied to. Yet, of course, there is never the method to detect phase transitions (and don’t claim that ours is). We saw that for our problem, i.e. for the model and data at hand, this might not be necessary and simple geometric analysis can be sufficient. One very interesting question is whether this is true for just this example, for some cases or maybe even a majority / all cases? The answer to this question is beyond the scope of our paper but an interesting one worth pointing out and investigating next. Having a situation in which our ML approach detects a phase transition and “no other existing approach works” would of course be interesting but probably very hard to find, if even possible. In our opinion, the main virtue of our approach is its generality and flexibility to adapt to the problem of interest by adjusting the network parameters. We benchmarked on a model that is well understood and demonstrated that it could be combined with state-of-the-art iPEPS simulations.

---

## Round 2 · List of Changes

- Added Fig. 4 and the corresponding paragraph (see author comment)
- We merged the duplicate reference 3 and 37 and added Ref. [6] at the appropriate position.
- We corrected y(x) to math mode.
- We changed “random initial states” to “random initial iPEPS” to make the distinction clear.
- We marked the term “anomaly detection” in italic at its first occasion in the text and abstract.
- We clarified the term "loss" at its first occurence.
- We added a footnote giving a short definition of training as data-specific optimization and refer to more details below in the text.
- We changed “In between, [..]” to “Between those states and the training region, [..]”.
- We thank the referee for pointing that out and updated Fig. 1 with increased training epochs.
- "Do the authors mean a representative _point_ in each phase? Otherwise, a little bit of physical intuition is being put in for the data generation." This is indeed misleading! We corrected it to “point”.
- In Fig. 3: We added the dotted lines indicating the theoretical transition and also added x-labels that we forgot in the earlier version.

---

## Editorial Decision

resubmitted